# Plantar threshold sensitivity assessment using an automated tool—Clinical assessment comparison between a control population without type 2 diabetes mellitus, and populations with type 2 diabetes mellitus, with and without neuropathy symptoms

Vitale Kyle Castellano[1][☯]*, Jon Commander[2,3][☯], Thomas Burch[1][☯], Hayden Burch[1][☯], Jessica Remy[3][☯], Benjamin Harman[3][☯], Michael E. Zabala[1][☯]

**1** Department of Mechanical Engineering, Samuel Ginn College of Engineering, Auburn University, Auburn, Alabama, United States of America, **2** Internal Medicine Associates, Opelika, Alabama, United States of America, **3** Edward via College of Osteopathic Medicine, Auburn, Alabama, United States of America

☯ These authors contributed equally to this work.
* vkc0005@auburn.edu

## Abstract

Diabetic peripheral neuropathy is often classified as a loss of sensation in the extremities, particularly in elderly populations. The most common diagnosis technique is with the use of the hand-applied Semmes-Weinstein monofilament. This study's first aim was to quantify and compare sensation on the plantar surface in healthy and type 2 diabetes mellitus populations with the standard Semmes-Weinstein hand-applied methodology and a tool that automates this approach. The second was to evaluate correlations between sensation and the subjects' medical characteristics. Sensation was quantified by both tools, at thirteen locations per foot, in three populations: Group 1-control subjects without type 2 diabetes, Group 2-subjects with type 2 diabetes and with neuropathy symptoms, and Group 3-subjects with type 2 diabetes without neuropathy symptoms. The percentage of locations sensitive to the hand-applied monofilament, yet insensitive to the automated tool was calculated. Linear regression analyses between sensation and the subject's age, body mass index, ankle brachial index, and hyperglycemia metrics were performed per group. ANOVAs determined differences between populations. Approximately 22.5% of locations assessed were sensitive to the hand-applied monofilament, yet insensitive to the automated tool. Age and sensation were only significantly correlated in Group 1 ($R^2 = 0.3422$, $P = 0.004$). Sensation was not significantly correlated with the other medical characteristics per group. Differences in sensation between the groups were not significant ($P = 0.063$). Caution is recommended when using hand-applied monofilaments. Group 1's sensation was correlated to age. The other medical characteristics failed to corelate with sensation, despite group.

**Data Availability Statement:** All relevant data are within the paper and its Supporting Information files.

**Funding:** This work was supported by the Edward Via College of Osteopathic Medicine [REAP Award 10307 FY19]. The design of the automated tool has been patented by the USPTO (US11426121B1). The funder did not impact analysis, decision to publish, or preparation of the manuscript in any way.

**Competing interests:** The authors have used the information from this study to inform the design of the automated tool, currently under a nonprovisional patent application (17027464). The patent application limits reporting transparency on the design of the automated tool. This does not alter our adherence to PLOS ONE policies on sharing data and materials.

# Introduction

Forty to sixty million people with diabetes experience complications due to peripheral neuropathy, globally [1]. Sensation loss in the extremities, such as the plantar and palmar surfaces, is often a complication attributed to those who have neuropathy [2]. Burning of the feet is the most common symptom linked to diabetes-related peripheral neuropathy (DPN), in addition to antalgic gait, and tingling [3–5]. Other symptoms range from insensitivity to temperature and accidental foot trauma, which can lead to infection, ulceration, and amputations [6,7]. Painful symptoms like burning, pins and needles, and even the sensation of electrical shock are prevalent in those who have neuropathy caused by diabetes, with nighttime being the time when symptoms are at their peak [8,9]. Aslam, Singh, and Rajbhandari have reported that hyperglycemia could be a factor attributed to painful symptoms, but other sources could be involved, such as damage to the nerves [8]. Bril reported that glycemic control is not a strong indicator for the development of diabetic neuropathy in persons with Type 2 diabetes mellitus (DM2), citing evidence that body mass index, high blood pressure, and even smoking could be additional factors [6]. In Yorek et al.'s review they reported that although glycemic control slowed down the progression of DPN in individuals with Type 1 diabetes, it did not in individuals with DM2 [10]. Noteworthy is that aging has been linked to the loss of sensation perception and affects 26% of individuals 65–74 years old and 54% of individuals 85 years of age and older [11].

Semmes-Weinstein monofilaments (SWM) are the most common method used to assess for sensation loss due to being inexpensive, fast, and noninvasive [12,13]. Developed by Semmes and Weinstein to replace the use of horsehair, these nylon monofilaments were used to assess for sensation loss on the palmer surfaces in subjects with brain injuries [13,14]. Monofilaments are applied normal to the surface of the skin until they buckle at which time they produce a constant force, the most common of which are calibrated to produce 10.0 gF [13,15,16]. There is a lack of consistent methodology with monofilament assessment for neuropathy, which can factor into the diagnosis of DPN [17]. In addition to a lack of consistency, the accuracy of the assessment is also affected by fatigue, angle and rate of insertion, application technique, and even the elasticity of the subject's skin [15,18].

Novel devices and techniques have been suggested to better measure degree of sensation loss, including a a rapid current threshold detection device (Neurometer®) and measuring an individual's oxygen level in their skin [19,20]. The work of Wilasrusmee et al. created a robot to apply a SWM at 10 gram-force (gF) to the plantar surface which agreed well with hand applied monofilament and vibratory assessments [17]. Similarly, Siddiqui et. al developed a robot which scanned the plantar surface through a clear medium with perforated holes, through which the device could apply a monofilament at 10.0 gF [21]. Spruce and Bowling developed a handheld electronic force sensor to mimic a typical monofilament; used an indicator light to tell the clinician when 10.0 gF had been applied upon contact [22]. This device not only had improved repeatability compared to the commercial monofilament but increased resistance to fatigue [22].

This study used an automated tool, similar to the robotic instruments previously mentioned, to determine a subject's plantar surface threshold sensitivity at several locations. The tool presented in this study can apply a variety of contact forces, consistent with SWM assessments, using a single monofilament, allowing for the mapping and documentation of the results per location. Therefore, the first aim of this study was to perform a comparison of the automated tool's findings with a standard hand applied 10.0 gF monofilament. The second aim of this study was to compare threshold sensitivity to age, body mass index (BMI), ankle brachial index (ABI), fasting blood sugar (FBS), and HbA1c in three populations: a control group without DM2, a DM2 group with neuropathy symptoms, and a DM2 group without

neuropathy symptoms. Studying how these demographics and medical characteristics relate to threshold sensitivity will reveal insights into how the presence of DM2 is related to an individual's degree of sensation loss.

## Materials and methods

### Populations and exclusion criteria

Volunteers were recruited in the Auburn-Opelika Alabama area. Research participants were not randomly recruited. Each subject volunteered themselves to be a part of this study, which makes this a based-on convenience sample. The study was conducted at Internal Medicine Associates (IMA) in Opelika, Alabama and the study was given IRB approval from the Edward Via College of Osteopathic Medicine (VCOM-Auburn, USA), under record number 2020–004. All subjects gave written consent to partake in this research study. All participants were required to have an ABI greater than or equal to 1.0 mmHg and had to be age forty or older. Four subjects failed the ABI screen and were excluded from the study. It should also be noted that none of the subjects had ulcers or a history of amputations on the plantar surface. Human subjects that passed the screening were categorized into three populations: Group 1. A control group without DM2 (n = 57, male = 29, female = 28), Group 2. A DM2 group with neuropathy symptoms (n = 58, male = 35, female = 23), and Group 3. A DM2 group without neuropathy symptoms (n = 38, male = 15, female = 23).

### Medical chart review and ABI assessment

The subjects having given their written consent to being a part of this study, were asked if they had DM2 and if so whether they had symptoms associated with neuropathy. Their responses then placed them into the appropriate group (1, 2, or 3). The medical profiles for each subject at IMA was reviewed for their age, BMI, FBS levels, and HbA1c levels and the data was recorded on a subject datasheet. The ABI for each subject was calculated and documented on their datasheet. If the subject's ABI was less than 1.0 mmHg, they would be dismissed from the study and would not be evaluated any further. Furthermore, FBS and HbA1c would both be averaged with up to the three most current readings. These average values were used for the analysis in this report. In Group 1, three individuals did not have FBS levels documented in their charts, while 27 individuals did not have HbA1c levels documented. However, these Group 1 subjects were still included in this study.

### Hand applied monofilament assessment

After collecting the subject's required medical data from chart review and measuring their ABI, a hand applied monofilament was used to assess for sensitivity on the plantar surfaces. The subject laid on the exam table, with their head elevated, and their feet were stretched out in front of them. Their calves were supported by the exam table, while their feet hung off the edge of the table with their toes pointing upward. A 10.0 gF hand applied SWM was used to evaluate for sensitivity at 26 locations, 13 locations per foot, Fig 1. The locations included the five toes (distal phalanges), five on the ball (distal metatarsals), and three on the heel (calcaneus). The monofilament was applied normal to each testing location by the clinician until it buckled. Subjects indicated whether they felt the hand applied stimulus per location, which was then documented in their datasheet by the clinician. When using the hand applied monofilament, the locations were assessed anterior to posterior, starting with the great toe. Following the great toe, the first distal metatarsal and then the first heel location were evaluated. This

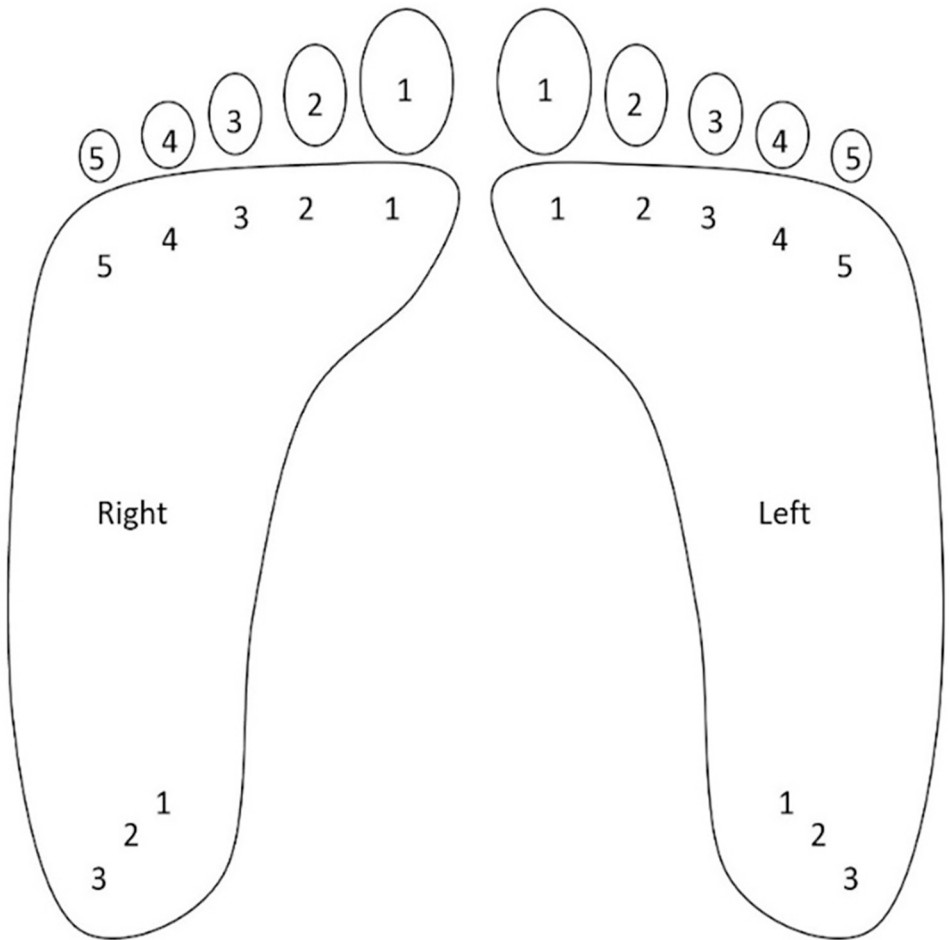

**Fig 1. Plantar surface assessment locations.** Thirteen locations per foot were evaluated with both a hand applied monofilament and the automated tool.

anterior to posterior procedure was performed at all five distal phalanges, five distal metatarsals, and three locations on the heel per foot.

## Automated tool assessment

Following the hand applied monofilament evaluation, an automated diagnostic tool, Fig 2A, was used on the same 13 locations one foot at a time. The automated tool was panted with the USPTO under US11426121B1. The diagnostic tool used a 10.0 gF rated monofilament which was mounted to a load cell feedback loop with a stepper motor, depicted in Fig 3. This probe subassembly's stepper motor (Walfront D8-MOTOR80 Stepper Motor) continues to actuate the monofilament against the skin until the load cell (RB-Phi-203 100g Micro Load Cell) measured the prescribed force. The Walfront stepper motor has a built-in lead screw driven carriage, with a pitch of 0.5 mm, equating to a linear travel distance of 0.025 mm per step. It also has 0.800 amps per phase and was 20 steps per revolution. Furthermore, the Walfront stepper motor was connected to a TB6600 stepper motor driver, which in turn was connected to an Arduino Uno. A miniature ball bearing carriage was used to stabilize the relative motion of the monofilament as it is driven forward and backward by the Walfront stepper motor. The loaded cell has a rated output of 600 μV/V and has a 1000-ohm impedance and was connected to a

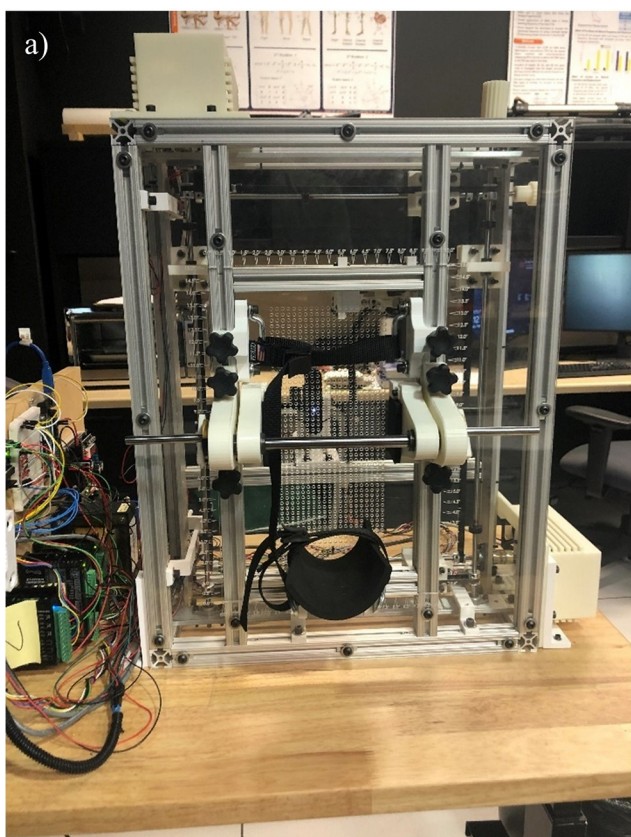 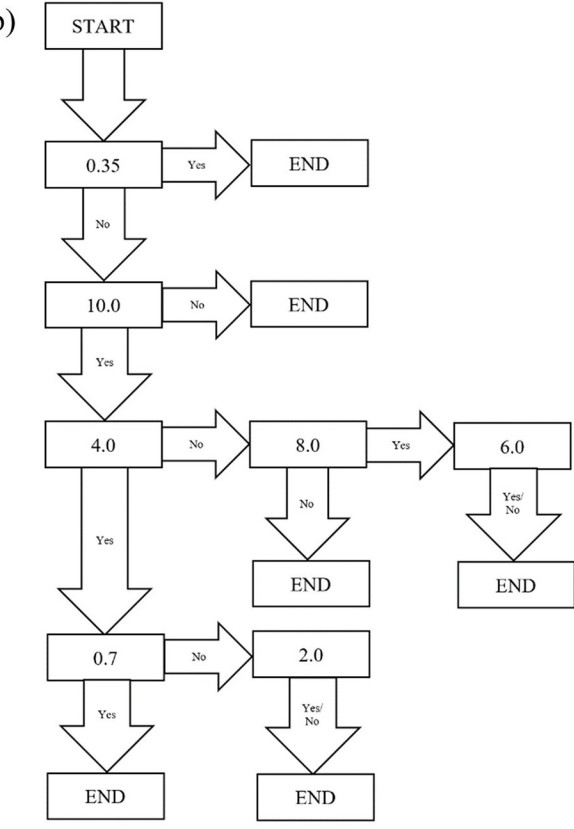

**Fig 2. Automated tool and homing sequence protocol.** a) Automated tool which applies a monofilament through the perforated foot plate until contact with the plantar surface, b) Homing sequence protocol used to determine threshold sensitivity per location. Units are in gF.

24-bit analog-to-digital HX711 amplifier, sampling data at 10 Hz. A homing sequence, Fig 2B, was used to determine the individual's threshold sensitivity at each location using the following forces: 0.35, 0.70, 2.0, 4.0, 6.0, 8.0, and 10.0 gF, which are typical in monofilament assessments [23]. The subject sat on the exam table, with their back tilted against the backrest, while their calves were supported by the exam table. The subject's foot was placed against a perforated foot plate made from clear acrylic, with 1,029 holes. This array of holes fit within a 5-inch by 7-inch rectangle and were evenly spaced apart by 0.25 inches. The device included a foot clamping mechanism, presented in Fig 4, which can be adjusted to fit various sizes of feet by sliding the components along the 80/20 aluminum extrusions. The clamping mechanism also provides light compression to the sides of the foot and allows the subjects foot to be relocated to an identical orientation for follow-up assessments. EVA foam was placed on all rigid surfaces that were in direct contact with the subject's skin. Their foot was secured in place with straps over the toes and over the ankle, with their toes pointing straight upward. Additionally, the chassis of the device was created using 17 additional 80/20 aluminum extrusions, as seen in Fig 5. After the subject's foot was secured in place a camera (Logitech Brio) mounted behind the device was used to take a photograph, which was then used in a MATLAB (MATLAB R2018a) script to allow the clinician to select testing locations for evaluation. The camera takes a picture through the clear acrylic foot plate, as depicted in Fig 6, which is then used by the machine operator to select locations for assessment. Testing locations were grouped by three regions in the automated tool assessment: toe region, ball region, and heel region. The

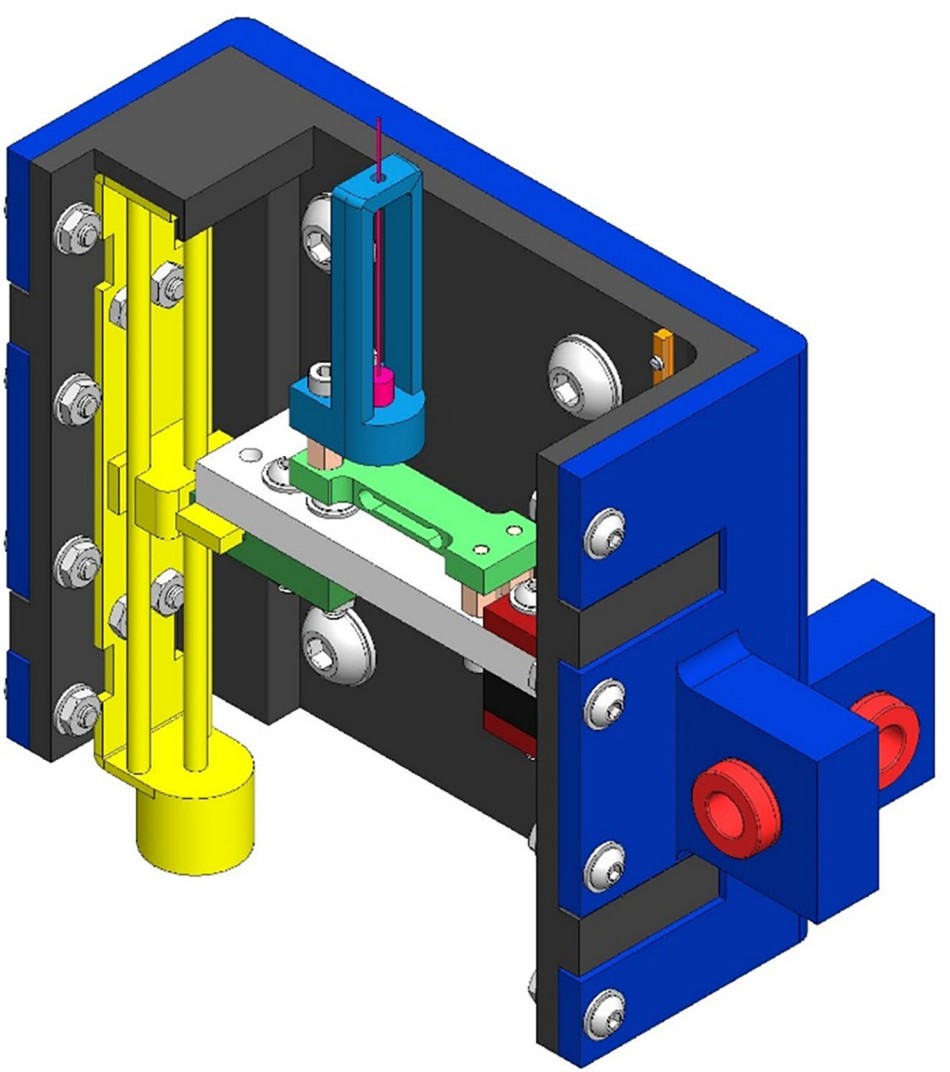

**Fig 3. Probe subassembly.** Monofilament (Magenta), Load Cell (Green), Stepper Motor (Yellow).

MATLAB script included randomization within regions. Furthermore, the MATLAB script communicated with Arduino microcontrollers (Arduino Uno and Arduino Mega) which controlled GT2 belts and pulleys used to translate the monofilament to each testing location. Specifically, the Arduino Mega communicated with the gantry subassembly, Fig 7, while the Arduino Uno communicated with the probe subassembly, Fig 3. The gantry subassembly used two stepper motors (STP-MTS-17040 Stepper Motor) to convert the rotation of the stepper motors to linear motion, resulting in the motion of the probe subassembly. These stepper motors are NEMA 17 bipolar stepper motors, with 3.81 lb-in of torque, 1.7 amps per phase, and have 200 steps per revolution. The gantry subassembly included linear sleeve bearings, linear mounted bearings, shaft collars, linear motion shafts, and 3D printed carriages to allow for reliable positioning of the probe subassembly. The shaft collars serve as physical limits for the machine. Limit switches mounted along the principal axes of the gantry subassembly are used for initial calibration of the system before the assessment. The device was powered by a STP-PWR-4808 48-volt, 5-amp, power supply and was controlled by a Dell Inspiron 15 3000

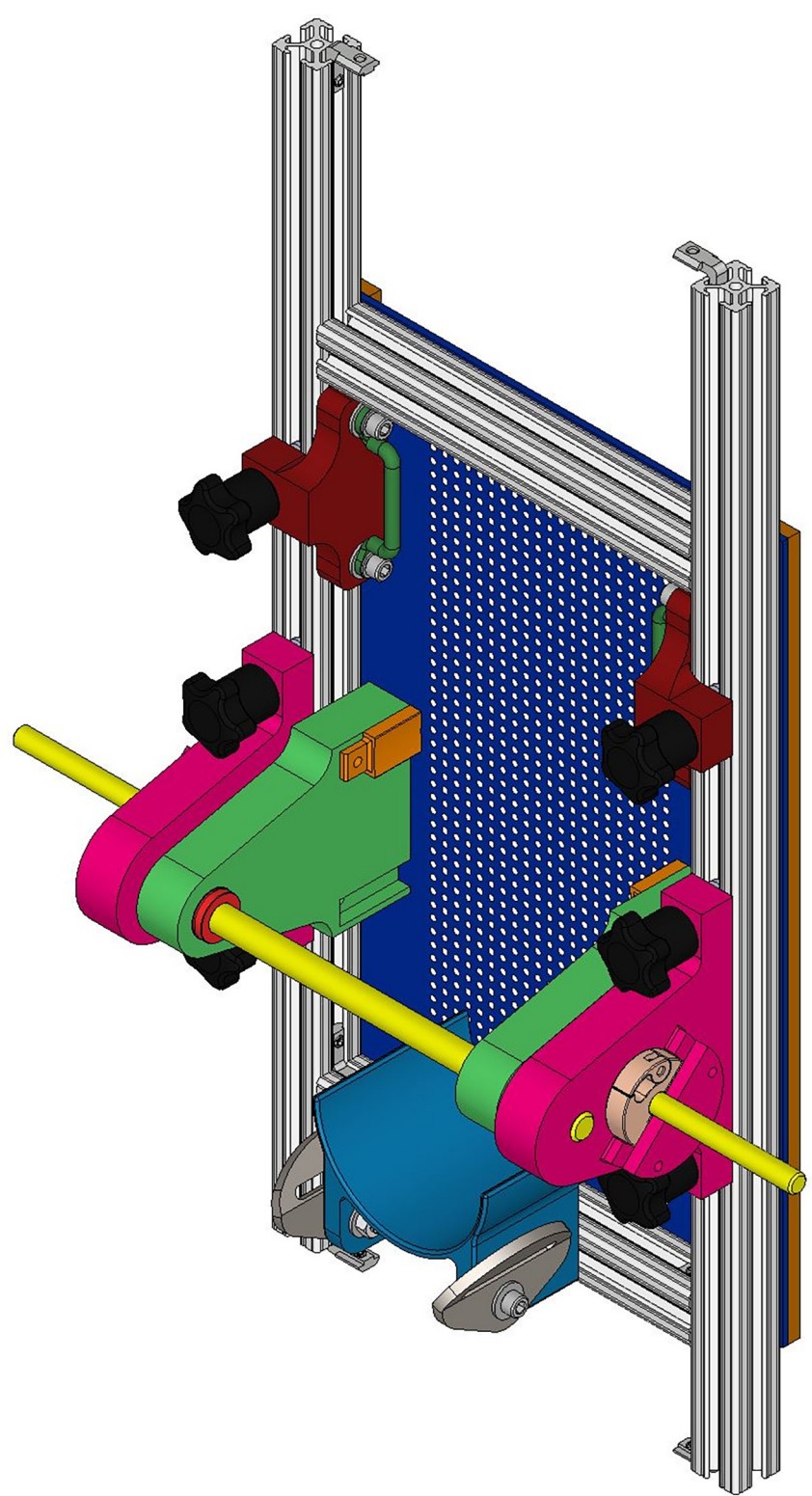

**Fig 4. Foot clamping mechanism.** Perforated Foot Plate (Dark Blue), Aluminum 80/20 Extrusions (White), Toe Clamp (Maroon), Foot Clamp Base (Magenta), Foot Clamp (Green), and Ankle Holder (Blue).

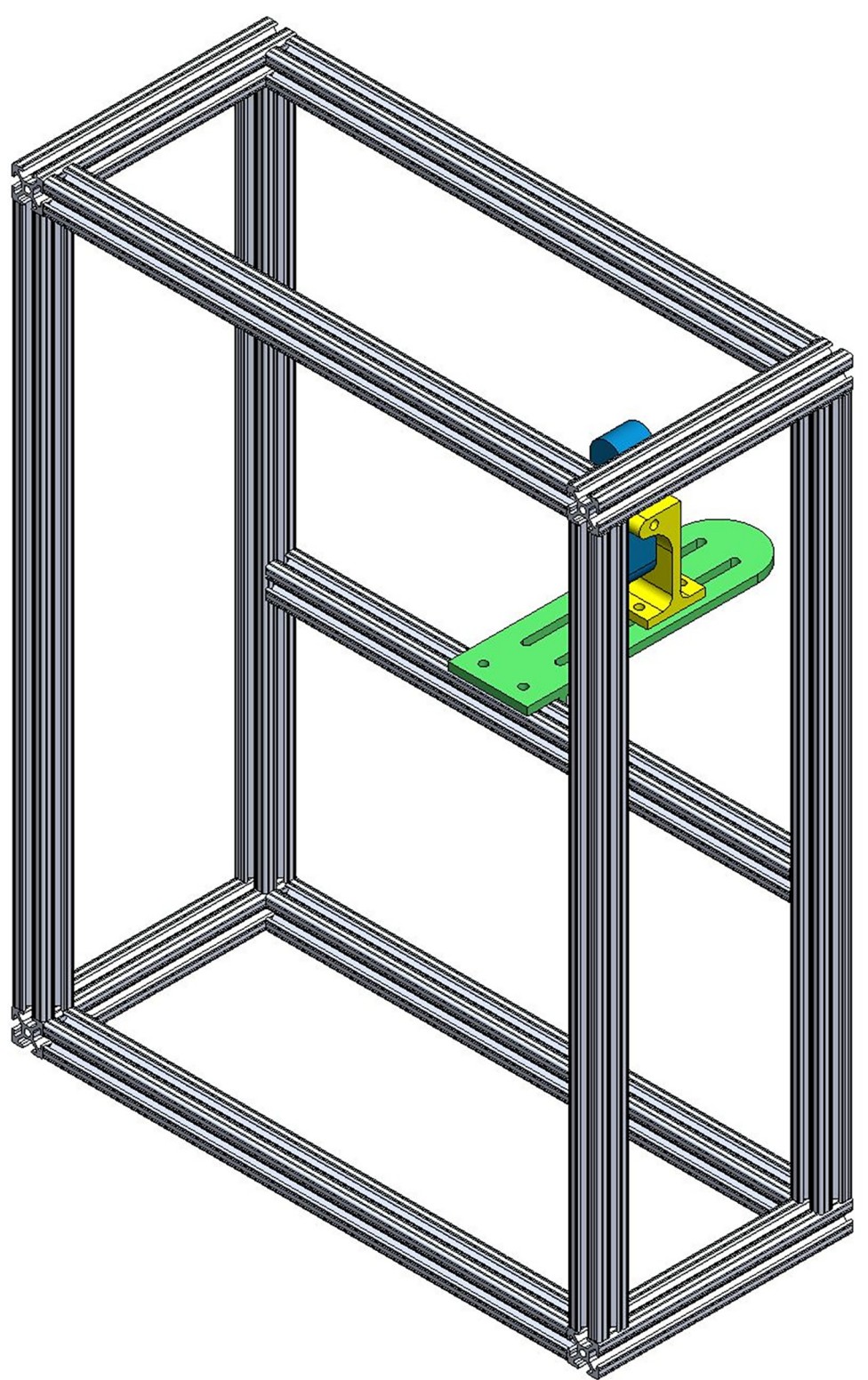

**Fig 5. Automated tool chassis.** Aluminum 80/20 Extrusions (White), Camera Mounting Plate Green), Camera Mount (Yellow), and Logitech Brio Webcam (Blue).

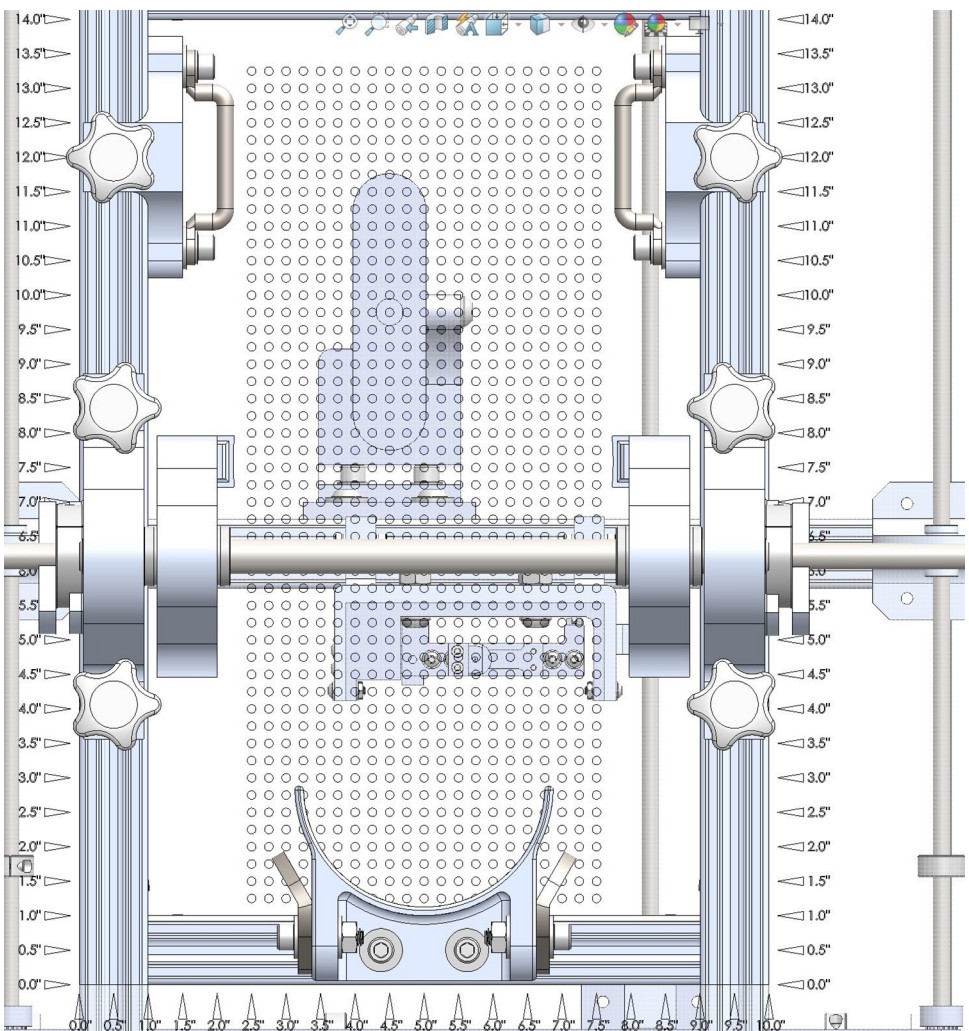

**Fig 6. Foot plate.** Acrylic perforated foot plate with Logitech Brio webcam and Probe Subassembly visible.

Touch Laptop. Overall, the device weighed approximately 45 pounds and had a physical footprint of 23.5 inches in width, 29 inches in height, and 19.75 inches in depth. The MATLAB functions and the Arduino Uno function used for this device are included in the S1 File of this manuscript. The Arduino Mega used a common g-code interpreter known as GRBL, which can be downloaded through GitHub. Once the monofilament was at the desired location it was actuated through the hole of the perforated foot plate until it contacted the skin at the prescribed force. After the monofilament was applied, subjects used a LED pushbutton to indicate their response to sensing the applied stimulus. Each time a stimulus was applied the LED pushbutton would start to blink, during which time the subject depressed the pushbutton for one second only if they felt the prescribed force. Subjects were given five seconds to catalogue their response. Also, false positive assessments were randomly incorporated into the automated tool evaluation, with each of the thirteen locations per foot having a 10% chance of occurrence. During a false positive check, the automated tool actuated the monofilament forward, mimicking the sound of a standard application of the monofilament, but it did not make contact. The LED pushbutton also blinked, at which time the subject depressed it if they perceived a

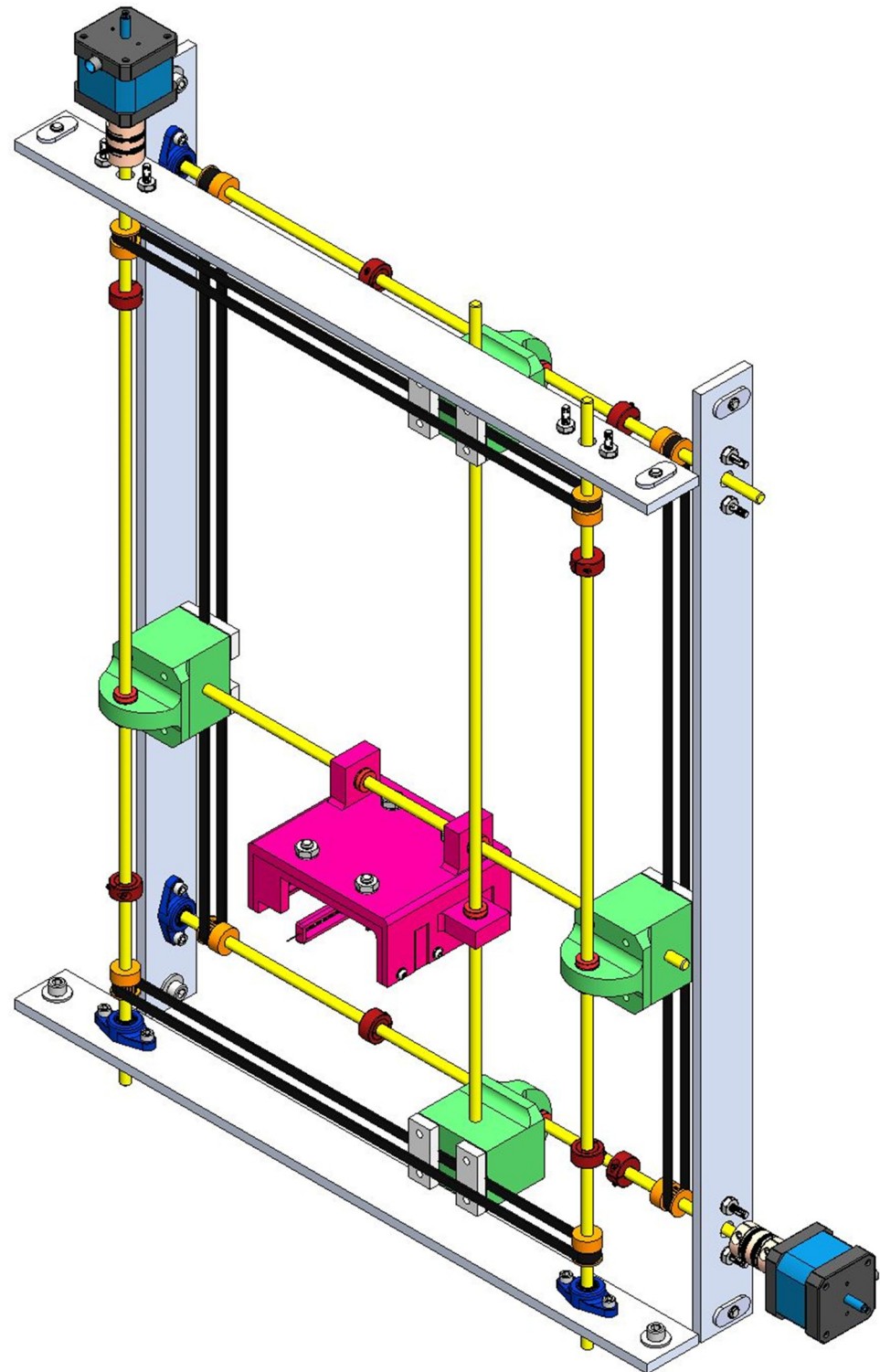

**Fig 7. Gantry subassembly.** Probe Subassembly (Magenta), Stepper Motor (Blue/Black), Belts (Black), Pulley (Orange), Linear Motion Rods (Yellow), Coupling (Bronze).

sensation, although one was not applied. At the end of the exam a threshold sensitivity map was generated using the MATLAB script, Fig 8. These procedures were repeated for the subject's second foot.

## Post processing of automated tool data

It was determined that subjects were firmly pressing their foot up against the foot plate, likely reducing the elasticity of their skin. This created an inaccuracy when the monofilament was applied, often resulting in a load greater than desired being applied. Although the load cell is constantly monitoring the contact force of the monofilament, the sensor's read rate prevents it from stopping the actuation of the monofilament instantaneously, especially when the subject's skin elasticity is significantly reduced. As such, every application of the monofilament via the automated tool was screened manually for accuracy in all force categories (0.35–10.0 gF). This was conducted after the subject completed the study. To ensure that accurate data was used it was decided that each location had to be within a 2.0 gF range for it to be accepted. However, at 0.35 gF and 0.70 gF, it was required that both needed to be less than 2.0 gF to be valid. If the actual force was outside of this range, then this application of the monofilament was a "device error". Although a "device error" lead to the removal of data for a specific application of the monofilament, it did not always correspond to a failure in determining an individual's threshold sensitivity at this location. Device errors also affected false positive assessments, as some subjects were deforming the footplate such that the monofilament made accidental contact, resulting in a load being applied. Locations where the threshold could be determined were assigned to the correct force classification: 0.35, 0.70, 2.0, 4.0, 6.0, 8.0, 10.0, and >10.0 gF. If a threshold sensitivity could not be determined due to the presence of "device errors" then this location was labeled as "indeterminate". Another instance where a location was labeled as "indeterminate" was if the wrong location was evaluated, which was caused by a camera misalignment. The photograph was realigned using MATLAB commands following the subject's evaluation, however if it was apparent that the wrong location was tested then it was excluded from future analysis.

## Threshold sensitivity calculation

Using the threshold sensitivities determined with the automated tool, a regional threshold sensitivity index (TSI) was calculated, which is an average of all the locations within the specific region, excluding indeterminate locations (Eq 1). Points were assigned to each threshold value (0.35 gF = 1, 0.70 gF = 2, 2.0 gF = 3, 4.0 gF = 4, 6.0 gF = 5, 8.0 gF = 6, 10.0 gF = 7, and >10.0 gF = 8). TSIs spanned one to eight, a score of one indicated a region with increased sensation perception, while a score of eight represented a region insensitive. Using all six regions per individual a TSI Norm was calculated to assign a single value representative of an individual's overall threshold sensitivity (Eq 2). It should be noted that TSI and TSI Norm are unitless quantities, attributed to these metrics being calculated using score-based averages.

$$TSI = \frac{0.35(1) + 0.70(2) + 2.0(3) + 4.0(4) + 6.0(5) + 8.0(6) + 10.0(7) + > 10.0(8)}{Locations - Indeterminate\ Locations} \quad (1)$$

$$TSI\ Norm = \sqrt{TSI_{R-Toe}{}^2 + TSI_{L-Toe}{}^2 + TSI_{R-Ball}{}^2 + TSI_{L-Ball}{}^2 + TSI_{R-Heel}{}^2 + TSI_{L-Heel}{}^2} \quad (2)$$

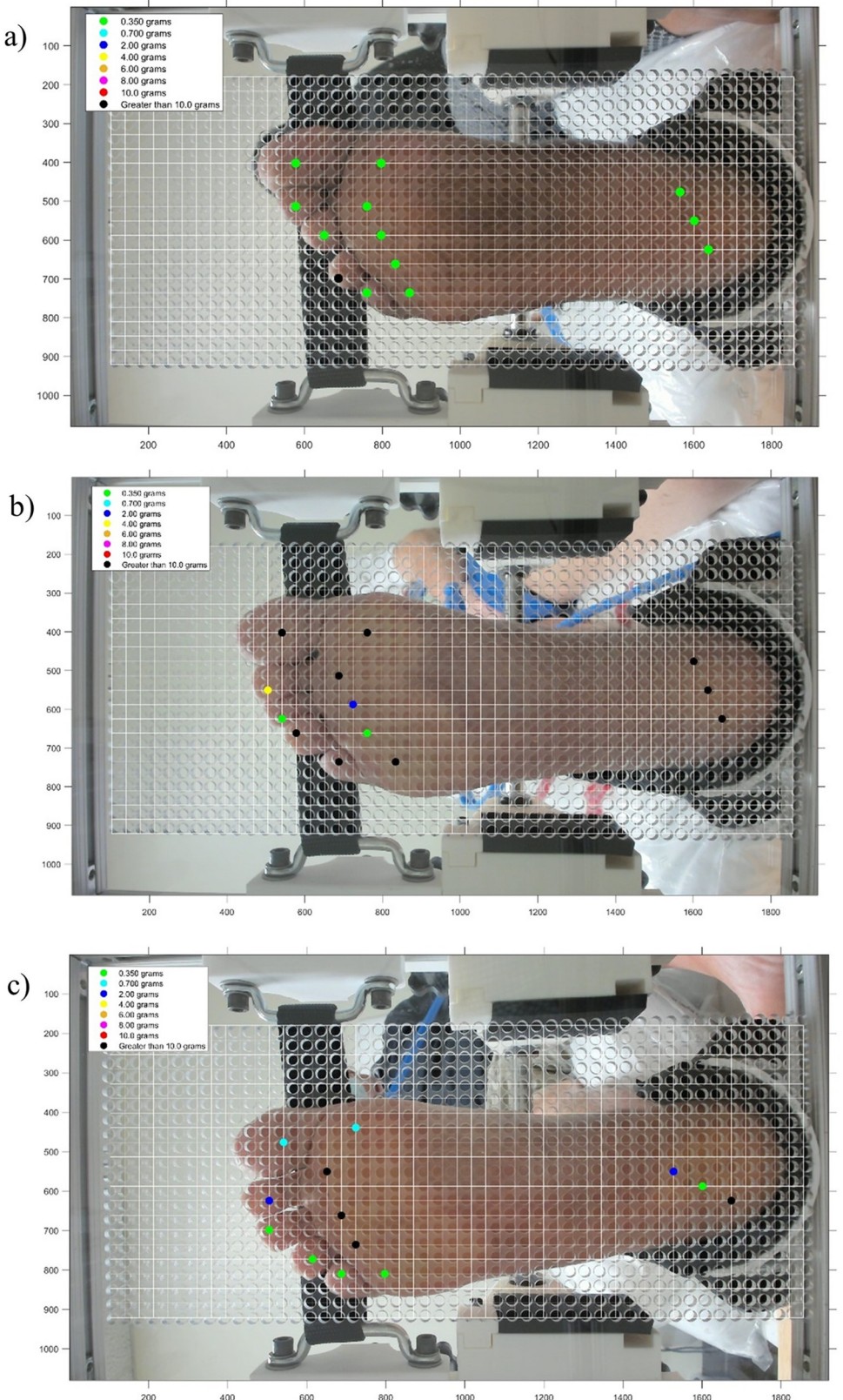

**Fig 8. Threshold sensitivity maps.** a) Subject from Group 1, b) Subject from Group 2, c) Subject from Group 3.

## Analysis approach

A comparative analysis between the hand applied monofilament and the automated tool was conducted for all testing locations. Percentages were calculated among all locations where the subjects stated that they felt the hand applied monofilament but were insensitive to the 10.0 gF applied by the automated tool. This percentage was considered the underdiagnosis rate of the hand applied monofilament at 10.0 gF and was calculated for all three groups.

The total number of false positive assessments were calculated per group and three percentages were calculated: fail rate, pass rate, and device error rate. The fail rate was the percentage of false positive check locations where the subject indicated that they felt a stimulus, though none was applied. The pass rate was the percentage of locations where subjects indicated that they did not feel a stimulus. The device error rate was the percentage of locations where the monofilament made accidental contact with the plantar surface.

## Statistical analyses

A linear regression analysis was performed between TSI Norm and the subject's age, BMI, ABI, FBS, and HbA1c in all three populations. Furthermore, to be a part of this analysis subjects had to have data at four out of five toes, four out of five ball, and two out of three heel locations per foot. This ensured that the presence of indeterminate locations did not skew the TSI Norms. As a result, 22 Group 1 subjects, 26 Group 2 subjects, and 22 Group 3 subjects were found to have enough valid testing locations per foot to be included in further analysis. Linear regression analysis was used to analyze the TSI Norm versus the other medical chart characteristics by using Microsoft Excel's regression data analysis tool, which outputted the $R^2$ and *P*-values. For regression analysis a *P*-value of 0.05 was significant. Next, R Studio (2022.02.0 Build 443) was used to conduct all statistical tests to find trends within each metric between the populations. TSI Norms and the other five medical characteristics were first checked for normal distribution and if the variance between populations were equal (homogeneity). Normality was assessed using visual inspection with Q-Q plots and the Shapiro-Wilk normality test ($P<0.05$). The data had to pass both tests to be considered normally distributed. If the data in each metric was found to be normally distributed, then a Brown-Forsyth equal variance assessment was used ($P<0.05$). If one or more of the groups were found to fail normality, then they were assessed for homogeneity using the Flinger-Killen test ($P<0.05$). If the data between all three populations were found to be both normal and with equal variances, then a one factor ANOVA was used to find statistical significance ($P<0.05$), followed by a Tukey post hoc assessment ($P<0.05$). Furthermore, a one factor ANOVA was also considered if the data lacked normality, but had homogeneity, given that ANOVA can handle some non-normality and that the populations were similar in number. If the data was normally distributed but lacked homogeneity then a Welch's ANOVA was used ($P<0.05$), followed by post hoc assessment with Pairwise Welch's t-test ($P<0.05$) with *P*-value adjustment using a Holm correction. If the data was not normal and did not have equal variances then a Permutational ANOVA was used to evaluate for significance ($P<0.05$), since it does not have any requirements for normality or homogeneity. This was followed by Pairwise Permutational t-Test with a P-value adjustment using a Holm correction ($P<0.05$). The Permutational ANOVA and its associated post hoc test was calculated using 10,000 iterations. Finally, Permutational ANOVA was also performed and then compared to the findings from the other ANOVA assessments. All relevant data are within this manuscript and its S1 File.

## Results

### Comparison between the hand applied monofilament and the automated tool

The 57 Group 1 subjects resulted in 1,185 locations assessed, after the removal of the indeterminate locations (n = 294). This yielded 21% of locations that were underdiagnosed. There were 1,334 locations analyzed, excluding the indeterminate locations (n = 172), within the 58 Group 2 subjects. Consequently only 24% of locations were underdiagnosed. Meanwhile 38 Group 3 subjects produced 870 testing sites, not including indeterminate locations (n = 118). Of these locations, 22% of them were underdiagnosed.

### False positive assessment results

In Group 1, 160 locations (11%) were subjected to a false positive check. Of these locations 13.75% of them failed the assessment, 57.5% of them passed, and 28.75% of them were device errors. In Group 2, 167 locations (11%) were evaluated with a false positive assessment. Of these 18% failed the test, 72% passed, while 10% experienced a device error. Finally in Group 3, 107 locations (11%) were assessed with a false positive. Of these locations, 19.6% of them failed, 72.0% of them passed, and 8.4% experienced a device error.

### TSI Norm and medical data outcomes compared to populations

**TSI Norm versus age.** The 22 Group 1 subjects demonstrated the strongest linear relationship between TSI Norm and age ($R^2$ = 0.3422, ***P* = 0.004**), which showed that as age increased, TSI Norm also increased, Fig 9A. Neither the 26 Group 2 subjects ($R^2$ = 0.0014, $P$ = 0.86) or the 22 Group 3 subjects ($R^2$ = 0.0083, $P$ = 0.69) yielded any meaningful relationship between TSI Norm and age. All groups passed both normality tests. As such a Brown-Forsyth test was used to assess for homogeneity between the groups, which found that the variances were barely

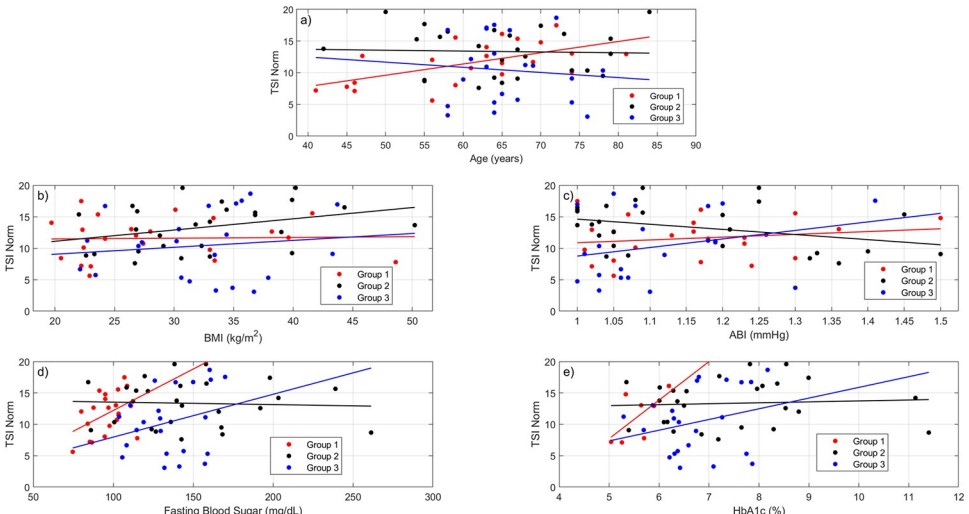

**Fig 9. Linear regression plots.** a) TSI Norm vs Age: Group 1 ($R^2$ = 0.3422, ***P* = 0.004**); Group 2 ($R^2$ = 0.0014, $P$ = 0.86); Group 3 ($R^2$ = 0.0083, $P$ = 0.69), b) TSI Norm vs BMI: Group 1 ($R^2$ = 0.0008, $P$ = 0.90); Group 2 ($R^2$ = 0.1144, $P$ = 0.091); Group 3 ($R^2$ = 0.0016, $P$ = 0.58), c) TSI Norm vs ABI: Group 1 ($R^2$ = 0.0305, $P$ = 0.44); Group 2 ($R^2$ = 0.1208, $P$ = 0.082); Group 3 ($R^2$ = 0.078, $P$ = 0.21), d) TSI Norm vs FBS: Group 1 ($R^2$ = 0.1819, $P$ = 0.054); Group 2 ($R^2$ = 0.0022, $P$ = 0.82); Group 3 ($R^2$ = 0.0692, $P$ = 0.24), e) TSI Norm vs HbA1c: Group 1 ($R^2$ = 0.4124, $P$ = 0.12); Group 2 ($R^2$ = 0.004, $P$ = 0.76); Group 3 ($R^2$ = 0.0628, $P$ = 0.26).

equal ($P$ = 0.052). An ANOVA was used to evaluate for significance ($P$ = 0.19), but since it was borderline homogenic a Welch's ANOVA was also used ($P$ = 0.21). Both of these and the Permutational ANOVA ($P$ = 0.20) found that the difference in mean ages between the populations was not significant (Group 1: 61.9±10.8 years, Group 2: 65.0±10.1 years, Group 3: 65.9±5.85 years). The results for age are categorized in Table 1.

**TSI Norm versus BMI.** TSI Norm was compared to BMI in all three groups, however none showed any type of strong linear relationship between these variables, Fig 9B. Group 1 ($R^2$ = 0.0008, $P$ = 0.90) and Group 3 ($R^2$ = 0.016, $P$ = 0.58) performed the worst, while Group 2 ($R^2$ = 0.1144, $P$ = 0.091) still did not yield a significant relationship with TSI Norm. Ultimately TSI Norm was insensitive to BMI. Group 1 did not pass the Shapiro-Wilk normality assessment (***P* = 0.014**), despite passing the visual inspection using a Q-Q plot, while Groups 2 and 3 passed both. A Flinger-Killeen variance assessment was used which found that the variances were equal between groups ($P$ = 0.70). An ANOVA was used to determine significance ($P$ = 0.11), in addition to a Permutational ANOVA ($P$ = 0.12). Both found that the mean BMIs between the populations were the same (Table 1).

**TSI Norm versus ABI.** TSI Norm as a function of ABI did not reveal any significant findings in the populations, Fig 9C. Group 1 ($R^2$ = 0.0305, $P$ = 0.44) and Group 3 ($R^2$ = 0.078, $P$-value = 0.21) showed the poorest linear relationship between TSI Norm and ABI. Group 2 ($R^2$ = 0.1208, $P$ = 0.082) yielded the best relationship but it was not significant. Furthermore, the ABIs for all three populations did pass visual inspection of the Q-Q plots, but Groups 2 and 3 were found to be nonnormal via the Shapiro-Wilk test. Flinger-Killeen test found that the populations had equal variance ($P$ = 0.34). The one-factor ANOVA and the Permutational ANOVA came to the same conclusion that the mean ABIs between the populations were not significantly different ($P$ = 0.56) (Table 1).

**TSI Norm versus FBS.** Linear regression analysis found that Group 1 showed the strongest relationship between TSI Norm and FBS ($R^2$ = 0.1819) but was barely insignificant ($P$ = 0.054), Fig 9D. It should be noted that out of the 22 Group 1 subjects, only 21 of them had FBS levels documented in their medical charts. The results of Group 2 ($R^2$ = 0.0022, $P$ = 0.82) and Group 3 ($R^2$ = 0.0692, $P$ = 0.24) did not yield a strong linear relationship. Normal distributions existed for all groups via Q-Q plot and Shapiro-Wilk normality assessments. The Brown-Forsyth homogeneity assessment found that the populations had unequal variances compared with each other (***P* = 0.0005**). The Welch's ANOVA assessment found that the difference in mean FBS between the groups was highly significant (***P* = 0.0005**) (Table 1). Group 1 was found to be significantly different to Group 2 (***P_{adj}*<0.0001**) and Group 3 (***P_{adj}*<0.0001**), using Pairwise Welch's t-Test. The DM2 groups when compared to each other were not significantly different ($P_{adj}$ = 0.19). The Permutational ANOVA concluded that the mean FBS was significantly different between groups (***P*<0.0001**). Pairwise Permutational t-Test found that Group 1 was significantly different to Group 2 (***P_{adj}* = 0.0003**) and to Group 3 (***P_{adj}* = 0.0003**). The post hoc test between the DM2 groups found that FBS was not significantly different between them ($P_{adj}$ = 0.19). The results for FBS are documented in Table 1.

**TSI Norm versus HbA1c.** Group 1 demonstrated the strongest relationship between TSI Norm and HbA1c ($R^2$ = 0.4124), but it lacked significance ($P$-value = 0.12), Fig 9E. However, Group 1 only had seven individuals out of the 22 who had HbA1c values documented in their medical charts. Group 2 ($R^2$ = 0.004, $P$-value = 0.76) and Group 3 ($R^2$ = 0.0628, $P$-value = 0.26) did not indicate a linear relationship between TSI Norm and HbA1c. All Q-Q plots passed the visual inspection, but Group 2 did not pass the Shapiro-Wilk test (***P* = 0.027**). This resulted in the use of the Flinger-Killeen variance assessment which found that the variances were unequal between the populations (***P* = 0.0015**). Due to nonnormality and unequal variances the Permutational ANOVA was the only ANOVA performed, which found that there were significant

**Table 1. Statistical analysis summary.**

| Metric | Group | Mean ± Standard Deviation | Q-Q Plots (Pass/Fail) | Shapiro-Wilk ($P<0.05$) | Brown-Forsyth ($P<0.05$) | Flinger-Killeen ($P<0.05$) | One-Factor ANOVA ($P<0.05$) | Welch's ANOVA ($P<0.05$) | Permutational ANOVA ($P<0.05$) | Pairwise Welch's t-Test ($P_{adj}<0.05$) | | | Pairwise Permutational t-Test ($P_{adj}<0.05$) | | |
|---|---|---|---|---|---|---|---|---|---|---|---|---|---|---|---|
| | | | **Normality Assessment** | | **Homogeneity of Variance Assessment** | | **ANOVA Assessment** | | | **Post Hoc Test** | | | | | |
| Age | | | | | 0.052 | | 0.19 | 0.21 | 0.20 | | | | | | |
| | Group 1 | 61.9±10.8 years | Pass | 0.43 | | | | | | | | | | | |
| | Group 2 | 65.0±10.1 years | Pass | 0.91 | | | | | | | | | | | |
| | Group 3 | 65.9±5.85 years | Pass | 0.12 | | | | | | | | | | | |
| BMI | | | | | | 0.70 | 0.11 | | 0.12 | | | | | | |
| | Group 1 | 28.6±7.78 kg/m$^2$ | Pass | **0.014** | | | | | | | | | | | |
| | Group 2 | 32.6±6.92 kg/m$^2$ | Pass | 0.42 | | | | | | | | | | | |
| | Group 3 | 32.1±6.01 kg/m$^2$ | Pass | 0.48 | | | | | | | | | | | |
| ABI | | | | | | 0.34 | 0.56 | | 0.56 | | | | | | |
| | Group 1 | 1.16±0.13 mmHg | Pass | 0.083 | | | | | | | | | | | |
| | Group 2 | 1.16±0.16 mmHg | Pass | **0.0027** | | | | | | | | | | | |
| | Group 3 | 1.12±0.11 mmHg | Pass | **0.022** | | | | | | | | | | | |
| FBS | | | | | **0.0005** | | | **<0.0001** | **<0.0001** | | A | B | | A | B |
| | Group 1 | 96.4±10.6 mg/dL | Pass | 0.93 | | | | | | | | | | | |
| | Group 2 | 148.1±43.4 mg/dL | Pass | 0.11 | | | | | | B | **<0.0001** | - | B | **0.0003** | - |
| | Group 3 | 135.5±20.1 mg/dL | Pass | 0.32 | | | | | | C | **<0.0001** | 0.19 | C | **0.0003** | 0.19 |
| HbA1c | | | | | | **0.0015** | | | **0.0025** | | | | | A | B |
| | Group 1 | 5.6±0.4% | Pass | 0.94 | | | | | | | | | | | |
| | Group 2 | 7.5±1.6% | Pass | **0.027** | | | | | | | | | B | **0.0006** | - |
| | Group 3 | 6.8±0.8% | Pass | 0.58 | | | | | | | | | C | **0.0006** | 0.053 |
| TSI Norm | | | | | **0.030** | | | 0.063 | **0.048***| | | | | A | B |
| | Group 1 | 11.6±3.28 | Pass | 0.78 | | | | | | | | | | | |
| | Group 2 | 13.3±3.65 | Pass | 0.15 | | | | | | | | | B | 0.17 | - |
| | Group 3 | 10.4±5.22 | Pass | 0.067 | | | | | | | | | C | 0.37 | 0.088 |

A-Group 1.

B-Group 2.

C-Group 3.

$P_{adj}$-Adjusted $P$-value.

*Permutational ANOVA average $P$-value after 10 iterations.

differences in the mean HbA1c (**P = 0.0025**) (Table 1). The Pairwise Permutational t-Test found that Group 1 was significantly different to Group 2 (**$P_{adj}$ = 0.0006**) and Group 3 (**$P_{adj}$ = 0.0006**). The DM2 groups were not significantly different from each other using this post hoc assessment, but it was close ($P_{adj}$ = 0.053). The HbA1c results are presented in Table 1.

**TSI Norm.** TSI Norms for all three groups could be approximated to follow a normal distribution using Q-Q plots and the Shapiro-Wilk normality assessment. However, the variances between the groups were unequal (**P = 0.030**), leading to the use of a Welch's ANOVA, which found that the mean TSI Norm between the populations was not significantly different (P = 0.063) (Group 1: 11.6±3.28, Group 2: 13.3±3.65, Group 3: 10.4±5.22). The Permutational ANOVA found a contrary finding, in which there was significance between the groups, however it should be noted that upon the first run of the Permutation ANOVA the P-value was equal to 0.0504 which is borderline significant. Every time a Permutational ANOVA is performed there are slight changes to its output due to the randomization aspect. As such 10 iterations of the Permutational ANOVA were performed and the average P-value was used (**P = 0.048**). Nonetheless post hoc assessment using the Pairwise Permutational t-Test did not find that any of the groups were significantly different to each other. Due to this it is more appropriate to say that the TSI Norms between the groups was not significant. Table 1 organizes the TSI Norm results accordingly.

## Discussion

### Hand applied monofilament and automated tool outcomes

The automated tool demonstrated that 21–24% of the locations assessed using the hand applied monofilament were underdiagnosed, regardless of cohort. This means that subjects screened for neuropathy using a 10.0 gF hand-applied monofilament could potentially be underdiagnosed for their current level of sensation. This combined with the previously discussed challenges with using hand applied monofilaments (fatigue, angle, and rate of insertion, etc.) implies that caution should be taken when relying on them to determine an individual's degree of sensation.

### False positive assessment outcomes

Of the locations assessed 11% of them received a false positive assessment, per group. In the methodology developed for the automated tool each location has a 10% chance of having a false positive check, which compared well to this observation. Between 13–20% of locations failed the false positive assessment. Groups 2 and 3 had the greatest pass rate (72%) and the lowest device error rates for false positive assessments (10% and 8.4%, respectively). Group 1 had the greatest amount of device error rate locations (28.75%), but also had the greatest number of total device errors (n = 294). This likely impacted the failure and pass rates for Group 1 locations.

### TSI Norm and medical data outcomes

The results showed that between the three groups, age, BMI, and ABI were not significantly different. It was also determined that the FBS and HbA1c between the groups were significantly different. This is not surprising considering that generally individuals with diabetes have elevated FBS and HbA1c, which is used as a criterion to assess for diabetes [24]. However, this study failed to find a significant difference between the three groups and TSI Norm (Table 1). Welch's ANOVA yielded an insignificant finding (P = 0.063), while the Permutational ANOVA found significance using a P-value averaged with 10 iterations (**P = 0.048**). Yet

the Pairwise Permutational t-Test post hoc assessment found that none of the comparisons were significant. It was concluded that TSI Norm was not significantly different between groups, which was not expected. Also surprising was that despite TSI Norm being dependent on age in Group 1 with a strong linear relationship ($R^2$ = 0.34, ***P* = 0.004**), it was independent in Groups 2 and 3. Meanwhile BMI, ABI, FBS, and HbA1c produced no significant findings within each group when compared to TSI Norm. Although in Group 1, TSI Norm and FBS yielded a strong linear relationship ($R^2$ = 0.19), it was barely insignificant ($P$ = 0.054). It is apparent that Group 1 subjects attributed their current degree of sensation to their age. Meanwhile DM2 subjects' degree of sensation was regardless of neuropathy symptoms. This matched well with the findings of Yorek et al., where glycemic control did not affect sensitivity [10]. The presence of DM2 in subjects makes predicting threshold sensitivity unlikely given the medical characteristics examined in this study, which highlights the necessity for not only neuropathy screening for those who currently do not express symptoms, but also continued assessment for subjects who have symptoms. The use of the automated tool developed does not only provide a more accurate assessment for threshold sensitivity but can also document the presence of an individual's degree of sensation over the course of multiple evaluations.

## Limitations

There are some limitations that need to be considered when interpreting the results of this study. The first was that all subjects were volunteers and were not chosen randomly from a larger population. This was an IRB requirement for the research study. Although it would have been ideal to randomly enroll patients that meet the requirements of this study it was not feasible given that the study only took place at one medical clinic. Future studies that consider subjects from more than one medical practice are recommended. Moreover, one of the biggest limitations with this study was that there were not enough participants in each of the three groups. Having access to 100 subjects per group would give greater statistical power when interpreting the results. It would have also been more effective to use random the testing order of the hand applied monofilament assessment, which would have been a better comparison to the randomized testing order performed by the automated tool.

## Clinical translations and significance

The automated tool has demonstrated an effective solution in determining the plantar threshold sensitivity of an individual by using a stepper motor load cell feedback loop to precisely measure the applied force. It can map a subjects threshold sensitivity on the bottom of their feet and can be used to monitor neuropathy progression over time. Unlike the current hand applied monofilament, this device can provide a range of contact forces to the plantar surface, which when paired with the documentation features of the device provide both the physician and the patient valuable insight into the degree of sensation loss present. The efficacy of treatments can also be objectively evaluated using the automated tool in future clinical studies. The automated tool not only allows for a more accurate assessment but provides the potential for the standardization of how neuropathy is measured. The use of randomizing testing order location and false positive assessments are invaluable to future studies that wish to conclude medical findings using the invention detailed in this manuscript.

This paper examined the relationship between sensation loss expressed as TSI Norm to age, BMI, ABI, FBS, and HbA1c within three different. Although age corelated to sensation loss in the Group 1 subjects, those who had DM2 did not corelate their sensation loss to any of the medical characteristics examined. As such the results indicated that it is more challenging to predict a DM2 subject's degree of sensation loss by just examining the medical characteristics

examined in this study. With this in mind it is recommended that those with DM2 should have regular screenings for neuropathy, regardless of the presence of symptoms. Future work should be focused on developing other tools and devices which can accurately measure sensation loss, which can then be used to validate the findings of this manuscript.

## Supporting information

**S1 Dataset. Medical results from study.**
(PDF)

**S1 File. MATLAB and Arduino code for automated tool.**
(PDF)

## Acknowledgments

The authors would like to thank the following individuals for their contributions towards the study: Kenny Brock, Yousef Nikzai, Austin Gould, Graham Trott, Nathan Anthony, David Axford, David McGregor, Chad Gibbs, Wesley Ortmann, Katie Allen, Bradley Louis, and Raydeer Piromari.

## Author Contributions

**Conceptualization:** Vitale Kyle Castellano, Jon Commander, Thomas Burch, Michael E. Zabala.

**Data curation:** Vitale Kyle Castellano.

**Formal analysis:** Vitale Kyle Castellano.

**Funding acquisition:** Jon Commander, Thomas Burch, Michael E. Zabala.

**Investigation:** Jessica Remy, Benjamin Harman.

**Methodology:** Vitale Kyle Castellano, Jon Commander.

**Project administration:** Jon Commander.

**Software:** Vitale Kyle Castellano, Hayden Burch.

**Supervision:** Jon Commander, Michael E. Zabala.

**Validation:** Vitale Kyle Castellano, Hayden Burch.

**Visualization:** Vitale Kyle Castellano.

**Writing – original draft:** Vitale Kyle Castellano.

**Writing – review & editing:** Vitale Kyle Castellano, Jon Commander, Thomas Burch, Hayden Burch, Jessica Remy, Benjamin Harman, Michael E. Zabala.

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
