## [Decision Letter · Decision Letter 0]

13 Oct 2022

PONE-D-22-20166Plantar threshold sensitivity assessment using an automated tool - Clinical assessment comparison between a control population without type 2 diabetes mellitus, and populations with type 2 diabetes mellitus, with and without neuropathy symptomsPLOS ONE

Dear Dr. Castellano,

Thank you for submitting your manuscript to PLOS ONE. After careful consideration, we feel that it has merit but does not fully meet PLOS ONE’s publication criteria as it currently stands. Therefore, we invite you to submit a revised version of the manuscript that addresses the points raised during the review process.

We look forward to receiving your revised manuscript.

Kind regards,

Yaodong Gu

Academic Editor

PLOS ONE

Journal Requirements:

2. We note that you have a patent relating to material pertinent to this article. Please provide an amended statement of Competing Interests to declare this patent (17027464), along with any other relevant declarations relating to employment, consultancy, patents, products in development or modified products etc. Please confirm that this does not alter your adherence to all PLOS ONE policies on sharing data and materials, as detailed online in our guide for authors http://journals.plos.org/plosone/s/competing-interests by including the following statement: "This does not alter our adherence to  PLOS ONE policies on sharing data and materials.” If there are restrictions on sharing of data and/or materials, please state these. Please note that we cannot proceed with consideration of your article until this information has been declared.

3. We noted in your submission details that a portion of your manuscript may have been presented or published elsewhere. [Yes, the clinical data from the control subjects was included in a publication currently in review in a different journal. The other paper was primarily concerned about device performance, but I did present the comparison between the control subject's sensitivity and their medical characteristics. However this paper compares the control subjects' results to those from type 2 diabetic groups and draws conclusions based on the differences between the groups. The paper being submitted to PLOS ONE does use similar calculations and methods from the other pending manuscript, but the statistical analysis is different and the goal of this paper is to draw conclusions about how diabetics affects threshold sensitivity, while the other paper is more focused on device performance.] 

Please clarify whether this publication was peer-reviewed and formally published. If this work was previously peer-reviewed and published, in the cover letter please provide the reason that this work does not constitute dual publication and should be included in the current manuscript.

Additional Editor Comments:

Please describe the sample selection criteria.

Reviewers' comments:

Reviewer's Responses to Questions

**Comments to the Author**

1. Is the manuscript technically sound, and do the data support the conclusions?

Reviewer #1: Partly

Reviewer #2: Yes

2. Has the statistical analysis been performed appropriately and rigorously? 

Reviewer #1: Yes

Reviewer #2: Yes

3. Have the authors made all data underlying the findings in their manuscript fully available?

Reviewer #1: Yes

Reviewer #2: Yes

4. Is the manuscript presented in an intelligible fashion and written in standard English?

Reviewer #1: No

Reviewer #2: Yes

5. Review Comments to the Author

Reviewer #1: Overall, this study addresses an important public health and Diabetes Complications Diabetic peripheral neuropathy is often classified as a loss of sensation in the extremities,

particularly in elderly populations. The most common diagnosis technique is with the use of the

hand-applied Semmes-Weinstein monofilament. This study’s first aim was to quantify and

compare sensation on the plantar surface in healthy and Type 2 diabetes mellitus (DM2)

populations with the standard Semmes-Weinstein hand-applied methodology and a tool that

automates this approach. The second was to evaluate correlations between sensation and the

subjects’ medical characteristics.

a) There is no mention of how the sample was selected and what type of sample was being selected (based on convenience sample?).

b) What are inclusion and exclusion criteria?

c) The response rate was not stated.

e) It seems subject participants are voluntary and non-randomly selected which subject and conclusion might be considered as a bias

Approximately 22.5% of locations assessed were sensitive to the hand-applied

monofilament, yet insensitive to the automated tool. Age and sensation were only significantly

correlated in Group 1 (R2=0.3422, P=0.004). Sensation was not significantly correlated with the

other medical characteristics per group. Differences in sensation between the groups were not

significant (P=0.063). Caution is recommended when using hand-applied monofilaments. Group

1’s sensation was correlated to age. The other medical characteristics failed to corelate with

sensation, despite group.

The authors should report some of the key limitations of this study in detailed,

I think the major concern of this submission is it lacks sufficient novelty and or original study, although, the study does not contribute novel knowledge or add sufficiently to the current literature, but, it would help local policy makers.

Reviewer #2: The paper "Plantar threshold sensitivity assessment using an automated tool - Clinical assessment comparison between a control population without type 2 diabetes mellitus, and populations with type 2 diabetes mellitus, with and without neuropathy symptoms" is carefully reviewed. Authors studied the role of an automated tool to determine plantar surface threshold sensitivity in diabetic and non-diabetic subjects.

The manuscript sound scientific and ethical enough. The novelty can not be ignored. However, there are few issues that should be revised.

- I recommend against use of abbreviations in first mention in the abstract and in the text (Several samples; BMI, ABI, and HbA1c).

- Methods expressed very well as did statistical analyses. However, I suggest moving statistics under a Statistical Analyses subheading.

- Results make sense and I think objective of the study were met. Tables are informative. Expression of significant p values as bold characters is advised.

- Discussion is too short. I recommend discussing similar works along with the results of the present study's results. Moreover, not only limitations, but also strengths of the present work should be mentioned. Lastly, comment on possible clinical translation of the study outcomes.

- Twelve of the references listed are older than 10 years. If appropriate, replace them with novel works, please.

6. PLOS authors have the option to publish the peer review history of their article (what does this mean?). If published, this will include your full peer review and any attached files.

Reviewer #1: **Yes: **Ptrof. Abdulbari BENER

Reviewer #2: **Yes: **Gulali Aktas

---

## [Author Response · Author response to Decision Letter 0]

25 Nov 2022

Comments from the Editor

• Comment 1: "Please ensure that your manuscript meets PLOS ONE's style requirements, including those for file naming."

Response: Thank you for providing the formatting guidelines for PLOS ONE and for the opportunity to submit revisions. The manuscript has been updated to reflect these requirements.

• Comment 2: "We note that you have a patent relating to material pertinent to this article. Please provide an amended statement of Competing Interests to declare this patent (17027464), along with any other relevant declarations relating to employment, consultancy, patents, products in development or modified products etc. Please confirm that this does not alter your adherence to all PLOS ONE policies on sharing data and materials, as detailed online in our guide for authors http://journals.plos.org/plosone/s/competing-interests by including the following statement: "This does not alter our adherence to PLOS ONE policies on sharing data and materials.” If there are restrictions on sharing of data and/or materials, please state these. Please note that we cannot proceed with consideration of your article until this information has been declared. This information should be included in your cover letter; we will change the online submission form on your behalf."

Response: Recently the USPTO has granted us a patent on the automated tool under patent US11426121B1. Therefore, this does not alter our adherence to PLOS ONE policies on sharing data and materials. We have updated our cover letter and our declaration of interest statements to reflect this, including the updated patent number. 

• Comment 3: "We noted in your submission details that a portion of your manuscript may have been presented or published elsewhere. [Yes, the clinical data from the control subjects was included in a publication currently in review in a different journal. The other paper was primarily concerned about device performance, but I did present the comparison between the control subject's sensitivity and their medical characteristics. However this paper compares the control subjects' results to those from type 2 diabetic groups and draws conclusions based on the differences between the groups. The paper being submitted to PLOS ONE does use similar calculations and methods from the other pending manuscript, but the statistical analysis is different and the goal of this paper is to draw conclusions about how diabetics affects threshold sensitivity, while the other paper is more focused on device performance.] Please clarify whether this publication was peer-reviewed and formally published. If this work was previously peer-reviewed and published, in the cover letter please provide the reason that this work does not constitute dual publication and should be included in the current manuscript."

Response: At this time the other manuscript is still in the review process at the other journal and has not been formally published. We have updated the cover letter to state this and why this other manuscript does not constitute a dual publication.

• Comment 4: "Please describe the sample selection criteria."

Response: The sample was based on convenience of the medical clinic where the research took place. Subjects were not randomly selected; each subject had to volunteer themselves to be a part of the study. The Populations and exclusion criteria section of the manuscript has been updated to make this clearer. 

Comments from Reviewer #1

• Comment 1: "There is no mention of how the sample was selected and what type of sample was being selected (based on convenience sample?)."

Response: The sample was based on convenience of the medical clinic where the research took place. Subjects were not randomly selected; each subject had to volunteer themselves to be a part of the study. The Populations and exclusion criteria section of the manuscript has been updated to make this clearer.

• Comment 2: "What are inclusion and exclusion criteria?"

Response: Thank you for your question. The inclusion and exclusion criteria were documented under the Populations and exclusion criteria subheading and the Medical chart review and ABI assessment subheading, both are located in the Materials and Methods of the manuscript. All participants had to have an ABI greater than or equal to 1.0 mmHg and needed to be forty years of age or older. The control subjects had to be non-diabetics. The diabetic groups were grouped based on the presence or absence of neuropathy symptoms. 

• Comment 3: "The response rate was not stated."

Response: Thank you for your comment. Since all subjects volunteered themselves and were not randomly selected from a larger sample there is not a response rate for this study. Our IRB prevented us from directly recruiting subjects. All subjects had to express their interest to participate. 

• Comment 4: "It seems subject participants are voluntary and non-randomly selected which subject and conclusion might be considered as a bias."

Response: Thank you for your insight on this point. This is correct, all subjects voluntarily participated in the study and were not randomly selected. In order to address the bias this poses on the study we have added a study limitations section in the discussion of this manuscript.

• Comment 5: "Approximately 22.5% of locations assessed were sensitive to the hand-applied

monofilament, yet insensitive to the automated tool. Age and sensation were only significantly

correlated in Group 1 (R2=0.3422, P=0.004). Sensation was not significantly correlated with the

other medical characteristics per group. Differences in sensation between the groups were not

significant (P=0.063). Caution is recommended when using hand-applied monofilaments. Group

1’s sensation was correlated to age. The other medical characteristics failed to corelate with

sensation, despite group.

The authors should report some of the key limitations of this study in detailed."

Response: Thank you for the recommendation on reporting the limitations of this study. We agree that this should be included in our manuscript. A dedicated limitations section has been added to the manuscript as a result of your feedback.

• Comment 6: "I think the major concern of this submission is it lacks sufficient novelty and or original study, although, the study does not contribute novel knowledge or add sufficiently to the current literature, but, it would help local policy makers."

Response: We would like to thank you for the time you have taken to review our manuscript and to provide valuable feedback. We feel that the device is novel, as it allows for the threshold sensitivity on the plantar surface to be mapped and documented. Although there are similar devices, there are currently no scholarly works that thoroughly depict how these similar devices work and what their limitations are. We feel that we have been very transparent on our automated tool, not only in terms of the design and how it functions but also in the results of the clinical study. The medical findings of our study have also been observed by other researchers, which we feel inadvertently validates this study.

Comments from Reviewer #2

• Comment 1: "I recommend against use of abbreviations in first mention in the abstract and in the text (Several samples; BMI, ABI, and HbA1c)."

Response: We would like to thank you for the time you have taken to review our manuscript and to offer your feedback. We have updated the abstract to remove the abbreviations. We used hyperglycemia metrics to remove HbA1c from the abstract. However, we feel that ABI, BMI, and HbA1c are very commonplace medical abbreviations and would prefer to leave them in the text and or figures/tables. 

• Comment 2: "Methods expressed very well as did statistical analyses. However, I suggest moving statistics under a Statistical Analyses subheading."

Response: Thank you for this suggestion, we agree that this is a beneficial change and have created a Statistical analyses subheading. 

• Comment 3: "Results make sense and I think objective of the study were met. Tables are informative. Expression of significant p values as bold characters is advised."

Response: The authors agree that bolding the significant p values is beneficial. This was carried out in the text, figure captions, and table 1.

• Comment 4: "Discussion is too short. I recommend discussing similar works along with the results of the present study's results. Moreover, not only limitations, but also strengths of the present work should be mentioned. Lastly, comment on possible clinical translation of the study outcomes."

Response: Thank you for your suggestion. We have updated our manuscript to include a limitations section, as well as a clinical translations and significance section. As of the time of this manuscript, there are no other works/manuscripts that are similar to this work. Although there are others who have invented similar automated tools for sensation loss evaluation, none of the other devices have been used in a clinical study similar to ours. Outside of other patents, there are some research articles on other inventions, but they are not as transparent with how their devices work and lacked the population size that we had. No other studies have used an automated tool that provides a range of forces to determine the threshold sensitivity for its subjects. We didn’t feel it was necessary to compare our device to the other inventions, since the focus of this paper was to use our automated tool to determine threshold sensitivity and then compare threshold sensitivity to medical characteristics. The goal of this paper was not to be a design paper, but rather a clinical assessment of our automated tool. We focused on comparing our medical results to those observed in other studies using different assessment techniques and general medicinal knowledge. 

• Comment 5: "Twelve of the references listed are older than 10 years. If appropriate, replace them with novel works, please."

Response: After reviewing the twelve references that were older than 10 years, we have removed six of them from our manuscript. We feel that the other references are still relevant to our paper and provided adequate background on neuropathy. We have cited similar devices in our introduction, but there are very few automated tools developed by other researchers. The ones that we did cite did not analyze sensation loss to medical characteristics.

---

## [Decision Letter · Decision Letter 1]

20 Mar 2023

PONE-D-22-20166R1Plantar threshold sensitivity assessment using an automated tool - Clinical assessment comparison between a control population without type 2 diabetes mellitus, and populations with type 2 diabetes mellitus, with and without neuropathy symptomsPLOS ONE

Dear Dr. Castellano,

Thank you for submitting your manuscript to PLOS ONE. After careful consideration, we feel that it has merit but does not fully meet PLOS ONE’s publication criteria as it currently stands. Therefore, we invite you to submit a revised version of the manuscript that addresses the points raised during the review process.

We look forward to receiving your revised manuscript.

Kind regards,

Hanna Landenmark

Staff Editor, PLOS ONE

on behalf of 

Yaodong Gu

Journal Requirements:

Additional Editor Comments:

Note from Staff Editor Hanna Landenmark (hlandenmark@plos.org): Please provide a completely transparent report of the design of the designed automated tool. We feel that without this, the manuscript is not fully reproducible as per PLOS ONE publication criterion number 3 (https://journals.plos.org/plosone/s/criteria-for-publication#loc-3).

Reviewers' comments:

Reviewer's Responses to Questions

**Comments to the Author**

1. If the authors have adequately addressed your comments raised in a previous round of review and you feel that this manuscript is now acceptable for publication, you may indicate that here to bypass the “Comments to the Author” section, enter your conflict of interest statement in the “Confidential to Editor” section, and submit your "Accept" recommendation.

Reviewer #1: All comments have been addressed

Reviewer #2: All comments have been addressed

2. Is the manuscript technically sound, and do the data support the conclusions?

Reviewer #1: (No Response)

Reviewer #2: Yes

3. Has the statistical analysis been performed appropriately and rigorously? 

Reviewer #1: Yes

Reviewer #2: Yes

4. Have the authors made all data underlying the findings in their manuscript fully available?

Reviewer #1: Yes

Reviewer #2: Yes

5. Is the manuscript presented in an intelligible fashion and written in standard English?

Reviewer #1: Yes

Reviewer #2: Yes

6. Review Comments to the Author

Reviewer #1: I am very pleased to confirm that the authors staisfactorily addressed and responded to my queries.

Reviewer #2: Thank you for addressing my comments appropriately. I recommend for publication of the paper in its current form.

7. PLOS authors have the option to publish the peer review history of their article (what does this mean?). If published, this will include your full peer review and any attached files.

Reviewer #1: **Yes: **Prof. Abdülbari BENER

Reviewer #2: **Yes: **Prof. Gulali Aktas

---

## [Author Response · Author response to Decision Letter 1]

26 Apr 2023

Comments from the Editor- Submitted April 2023

• Comment 1: Please review your reference list to ensure that it is complete and correct. If you have cited papers that have been retracted, please include the rationale for doing so in the manuscript text, or remove these references and replace them with relevant current references. Any changes to the reference list should be mentioned in the rebuttal letter that accompanies your revised manuscript. If you need to cite a retracted article, indicate the article’s retracted status in the References list and also include a citation and full reference for the retraction notice.

Response: Thank you for bringing this to our attention. We have verified that all references are complete and correct and that the Works cited is up to date. The second reviewer requested that we remove all sources older than 10 years, however after careful consideration we only removed 6 of the 12 that were older than 10 years. These sources were validated by sources that were already present in the manuscript and therefore replacements were not needed. The other 6 were left in because they remained relevant and are accurate based on our research and our collective experiences. We did not cite any papers that have been retracted. 

• Comment 2: Note from Staff Editor Hanna Landenmark (hlandenmark@plos.org): Please provide a completely transparent report of the design of the designed automated tool. We feel that without this, the manuscript is not fully reproducible as per PLOS ONE publication criterion number 3 (https://journals.plos.org/plosone/s/criteria-for-publication#loc-3).

Response: Thank you Hanna Landenmark for your time in reviewing our revised manuscript and for responding to my emailing regarding your suggestion. We have added in more details that fully disclosures the automated tool and makes it reproducible. We added specific specifications on the electronics used and added in more supporting details on how the machine operators. Additional figures accompanied these revisions. Previously we have provided all MATLAB and Arduino codes needed to operate the device. We hope these additions are satisfactory for publication in PLOS ONE, which we are very excited and grateful to be in. 

Comments from the Editor- Submitted November 2022

• Comment 1: Please ensure that your manuscript meets PLOS ONE's style requirements, including those for file naming.

Response: Thank you for providing the formatting guidelines for PLOS ONE and for the opportunity to submit revisions. The manuscript has been updated to reflect these requirements.

• Comment 2: We note that you have a patent relating to material pertinent to this article. Please provide an amended statement of Competing Interests to declare this patent (17027464), along with any other relevant declarations relating to employment, consultancy, patents, products in development or modified products etc. Please confirm that this does not alter your adherence to all PLOS ONE policies on sharing data and materials, as detailed online in our guide for authors http://journals.plos.org/plosone/s/competing-interests by including the following statement: "This does not alter our adherence to PLOS ONE policies on sharing data and materials.” If there are restrictions on sharing of data and/or materials, please state these. Please note that we cannot proceed with consideration of your article until this information has been declared. This information should be included in your cover letter; we will change the online submission form on your behalf.

Response: Recently the USPTO has granted us a patent on the automated tool under patent US11426121B1. Therefore, this does not alter our adherence to PLOS ONE policies on sharing data and materials. We have updated our cover letter and our declaration of interest statements to reflect this, including the updated patent number. 

• Comment 3: We noted in your submission details that a portion of your manuscript may have been presented or published elsewhere. [Yes, the clinical data from the control subjects was included in a publication currently in review in a different journal. The other paper was primarily concerned about device performance, but I did present the comparison between the control subject's sensitivity and their medical characteristics. However this paper compares the control subjects' results to those from type 2 diabetic groups and draws conclusions based on the differences between the groups. The paper being submitted to PLOS ONE does use similar calculations and methods from the other pending manuscript, but the statistical analysis is different and the goal of this paper is to draw conclusions about how diabetics affects threshold sensitivity, while the other paper is more focused on device performance.] Please clarify whether this publication was peer-reviewed and formally published. If this work was previously peer-reviewed and published, in the cover letter please provide the reason that this work does not constitute dual publication and should be included in the current manuscript.

Response: At this time the other manuscript is still in the review process at the other journal and has not been formally published. We have updated the cover letter to state this and why this other manuscript does not constitute a dual publication.

• Comment 4: Please describe the sample selection criteria.

Response: The sample was based on convenience of the medical clinic where the research took place. Subjects were not randomly selected; each subject had to volunteer themselves to be a part of the study. The Populations and exclusion criteria section of the manuscript has been updated to make this clearer. 

Comments from Reviewer #1- Submitted November 2022

• Comment 1: There is no mention of how the sample was selected and what type of sample was being selected (based on convenience sample?).

Response: The sample was based on convenience of the medical clinic where the research took place. Subjects were not randomly selected; each subject had to volunteer themselves to be a part of the study. The Populations and exclusion criteria section of the manuscript has been updated to make this clearer.

• Comment 2: What are inclusion and exclusion criteria?

Response: Thank you for your question. The inclusion and exclusion criteria were documented under the Populations and exclusion criteria subheading and the Medical chart review and ABI assessment subheading, both are located in the Materials and Methods of the manuscript. All participants had to have an ABI greater than or equal to 1.0 mmHg and needed to be forty years of age or older. The control subjects had to be non-diabetics. The diabetic groups were grouped based on the presence or absence of neuropathy symptoms. 

• Comment 3: The response rate was not stated.

Response: Thank you for your comment. Since all subjects volunteered themselves and were not randomly selected from a larger sample there is not a response rate for this study. Our IRB prevented us from directly recruiting subjects. All subjects had to express their interest to participate. 

• Comment 4: It seems subject participants are voluntary and non-randomly selected which subject and conclusion might be considered as a bias.

Response: Thank you for your insight on this point. This is correct, all subjects voluntarily participated in the study and were not randomly selected. In order to address the bias this poses on the study we have added a study limitations section in the discussion of this manuscript.

• Comment 5: Approximately 22.5% of locations assessed were sensitive to the hand-applied

monofilament, yet insensitive to the automated tool. Age and sensation were only significantly

correlated in Group 1 (R2=0.3422, P=0.004). Sensation was not significantly correlated with the

other medical characteristics per group. Differences in sensation between the groups were not

significant (P=0.063). Caution is recommended when using hand-applied monofilaments. Group

1’s sensation was correlated to age. The other medical characteristics failed to corelate with

sensation, despite group.

The authors should report some of the key limitations of this study in detailed.

Response: Thank you for the recommendation on reporting the limitations of this study. We agree that this should be included in our manuscript. A dedicated limitations section has been added to the manuscript as a result of your feedback.

• Comment 6: I think the major concern of this submission is it lacks sufficient novelty and or original study, although, the study does not contribute novel knowledge or add sufficiently to the current literature, but, it would help local policy makers.

Response: We would like to thank you for the time you have taken to review our manuscript and to provide valuable feedback. We feel that the device is novel, as it allows for the threshold sensitivity on the plantar surface to be mapped and documented. Although there are similar devices, there are currently no scholarly works that thoroughly depict how these similar devices work and what their limitations are. We feel that we have been very transparent on our automated tool, not only in terms of the design and how it functions but also in the results of the clinical study. The medical findings of our study have also been observed by other researchers, which we feel inadvertently validates this study.

Comments from Reviewer #2- Submitted November 2022

• Comment 1: I recommend against use of abbreviations in first mention in the abstract and in the text (Several samples; BMI, ABI, and HbA1c).

Response: We would like to thank you for the time you have taken to review our manuscript and to offer your feedback. We have updated the abstract to remove the abbreviations. We used hyperglycemia metrics to remove HbA1c from the abstract. However, we feel that ABI, BMI, and HbA1c are very commonplace medical abbreviations and would prefer to leave them in the text and or figures/tables. 

• Comment 2: Methods expressed very well as did statistical analyses. However, I suggest moving statistics under a Statistical Analyses subheading.

Response: Thank you for this suggestion, we agree that this is a beneficial change and have created a Statistical analyses subheading. 

• Comment 3: Results make sense and I think objective of the study were met. Tables are informative. Expression of significant p values as bold characters is advised.

Response: The authors agree that bolding the significant p values is beneficial. This was carried out in the text, figure captions, and table 1.

• Comment 4: Discussion is too short. I recommend discussing similar works along with the results of the present study's results. Moreover, not only limitations, but also strengths of the present work should be mentioned. Lastly, comment on possible clinical translation of the study outcomes.

Response: Thank you for your suggestion. We have updated our manuscript to include a limitations section, as well as a clinical translations and significance section. As of the time of this manuscript, there are no other works/manuscripts that are similar to this work. Although there are others who have invented similar automated tools for sensation loss evaluation, none of the other devices have been used in a clinical study similar to ours. Outside of other patents, there are some research articles on other inventions, but they are not as transparent with how their devices work and lacked the population size that we had. No other studies have used an automated tool that provides a range of forces to determine the threshold sensitivity for its subjects. We didn’t feel it was necessary to compare our device to the other inventions, since the focus of this paper was to use our automated tool to determine threshold sensitivity and then compare threshold sensitivity to medical characteristics. The goal of this paper was not to be a design paper, but rather a clinical assessment of our automated tool. We focused on comparing our medical results to those observed in other studies using different assessment techniques and general medicinal knowledge. 

• Comment 5: Twelve of the references listed are older than 10 years. If appropriate, replace them with novel works, please.

Response: After reviewing the twelve references that were older than 10 years, we have removed six of them from our manuscript. We feel that the other references are still relevant to our paper and provided adequate background on neuropathy. We have cited similar devices in our introduction, but there are very few automated tools developed by other researchers. The ones that we did cite did not analyze sensation loss to medical characteristics.

---

## [Editor Report · Decision Letter 2]

19 May 2023

Plantar threshold sensitivity assessment using an automated tool - Clinical assessment comparison between a control population without type 2 diabetes mellitus, and populations with type 2 diabetes mellitus, with and without neuropathy symptoms

PONE-D-22-20166R2

Dear Dr. Castellano,

We’re pleased to inform you that your manuscript has been judged scientifically suitable for publication and will be formally accepted for publication once it meets all outstanding technical requirements.

Kind regards,

Yaodong Gu

Academic Editor

PLOS ONE

Additional Editor Comments (optional):

The authors have done a good revision.
---

## [Editor Report · Acceptance letter]

29 Jun 2023

PONE-D-22-20166R2 

Plantar threshold sensitivity assessment using an automated tool - Clinical assessment comparison between a control population without type 2 diabetes mellitus, and populations with type 2 diabetes mellitus, with and without neuropathy symptoms 

Dear Dr. Castellano:

I'm pleased to inform you that your manuscript has been deemed suitable for publication in PLOS ONE. Congratulations! Your manuscript is now with our production department. 

Kind regards, 

on behalf of

Professor Yaodong Gu 

Academic Editor

PLOS ONE